# The Role of Purinergic Receptors in the Circadian System

**DOI:** 10.3390/ijms21103423

**Published:** 2020-05-12

**Authors:** Amira A.H. Ali, Gayaneh Avanes Avakian, Charlotte Von Gall

**Affiliations:** Institute of Anatomy II, Medical Faculty, Heinrich-Heine-University, Moorenstrasse 5, 40225 Düsseldorf, Germany; amira.ali@med.uni-duesseldorf.de (A.A.H.A.); GayanehAvakian@aol.com (G.A.A.)

**Keywords:** purinergic receptors, circadian clock, SCN, urinary bladder

## Abstract

The circadian system is an internal time-keeping system that synchronizes the behavior and physiology of an organism to the 24 h solar day. The master circadian clock, the suprachiasmatic nucleus (SCN), resides in the hypothalamus. It receives information about the environmental light/dark conditions through the eyes and orchestrates peripheral oscillators. Purinergic signaling is mediated by extracellular purines and pyrimidines that bind to purinergic receptors and regulate multiple body functions. In this review, we highlight the interaction between the circadian system and purinergic signaling to provide a better understanding of rhythmic body functions under physiological and pathological conditions.

## 1. Introduction

Purinergic signaling has been implicated in multiple brain functions such as learning and memory, locomotor and feeding behavior, and sleep (reviewed in [1]), as well as body functions such as gastrointestinal and cardiac functions or micturition (reviewed in [2,3]). Importantly, all of these brain and body functions show time-of-day-dependent variations controlled by the circadian system. Thus, in this review, we will summarize the knowledge on purinergic signaling within the mammalian circadian system.

## 2. The Circadian System and the Molecular Clockwork

Life on Earth has evolved under the influence of rhythmic changes in the environment. Thus, living organisms have developed internal circadian clocks, which allow anticipating these rhythmic changes and adapting their behavior and physiology accordingly. Circadian clocks continue to oscillate with a period length close to the solar day of approximately 24 h, even in the absence of a rhythmic light–dark cycle. Molecular clockwork ensures the precision. Under natural conditions, the phase and period of circadian clocks are entrained to the environmental time. In addition to persistence and resetting, a true circadian oscillator is temperature compensated. Circadian clocks consist of three major components: a central circadian oscillator, input pathways to allow entrainment, and output pathways that orchestrate circadian rhythms in behavior and physiology [4]. The circadian clock is found in various species. Cyanobacteria are the simplest organisms and the only prokaryotes known to have a robust circadian clock [5]. The cyanobacterial molecular clockwork is based on post-translational modification, and its main component KaiC is an autokinase, autophosphatase, and ATPase whose daily rhythms of phosphorylation and ATPase activity are key features of the timekeeping mechanism [5]. In eukaryotes, the molecular clock is based on autoregulatory transcription–translation feedback loops (TTFL) of so-called clock genes. The Nobel Prize in Physiology or Medicine 2017 was awarded jointly to Jeffrey C. Hall, Michael Rosbash, and Michael W. Young for their discoveries of molecular mechanisms controlling the circadian rhythm using *Drosophila* as a model organism. The major principle of the *Drosophila* molecular clockwork is conserved also in mammals. Here, the positive components are the transcription factors Clock and Bmal1 that heterodimerize and bind to E-box elements. The negative components are the periods (Per1 and Per2) and cryptochromes (Cry1 and Cry2) that form together with casein kinases, which regulate Per phosphorylation and turnover [6], a negative regulatory complex that inhibits Clock/Bmal1-dependent transcription [7]. Importantly, E-box elements are present in the regulatory regions, promoters of the genes encoding for the Pers and Crys, as well as in other genes encoding for key regulators in cell function; therefore, they represent the so-called clock-controlled genes (Ccg; Figure 1). Thus, in addition to a time-keeping mechanism, the molecular clockwork drives rhythmic cell function.

In mammals, the circadian system comprises the retina (light input), the hypothalamic suprachiasmatic nucleus (SCN; central circadian oscillator), and subordinate/peripheral oscillators in the brain and the body (Figure 2) [4]. Molecular clockwork is not only present in the SCN but also in subordinate oscillators in the brain and in the periphery. The SCN is directly entrained by light via input from retinal ganglion cells (RGCs). The axons of RGCs convey light/dark information to the SCN by the release of glutamate and pituitary adenylate cyclase-activating peptide (PACAP) ([8], reviewed in [9]). Subsequent signal transduction pathways including phosphorylation of the transcription factor CREB [10,11,12] and activation of *Per* expression leads to an adjustment of the molecular SCN clockwork (Figure 1). Moreover, the SCN provides rhythmic signals for the temporal synchronization of various peripheral organs and systems within the body. Important rhythmic signaling molecules within the circadian system are the “hormone of darkness” melatonin, which is released from the pineal gland and the retina (reviewed by [13,14]), and the “stress hormone” corticosterone, which is released from the adrenal gland [15,16]. In addition, the autonomic nervous system is under the control of the circadian system and controls rhythmic organ function [17]. This internal temporal synchronization is essential for mental and global health [18].

## 3. Purinergic Signaling

The purinergic signaling pathway is mediated by extracellular purines (adenosine, ADP, and ATP) and pyrimidines (UDP and UTP) through binding to purinergic receptors. Purinergic receptors are classified into P1 purinoreceptors that are stimulated by adenosine and P2 purinoreceptors, which are stimulated by variety of nucleotides. P1 receptors include A1, A2A, A2B, and A3 receptor subtypes, whereas the P2 receptors are further subdivided into G-protein-coupled receptors (P2Y) and ion channel ligand-gated receptors (P2X). P2Y receptors involve P2Y1, P2Y2, P2Y4, P2Y6, P2Y11, P2Y12, P2Y13, and P2Y14, while P2X are subclassified into P2X1–P2X7 subtypes [19,20].

Purinergic receptors can be found in almost every mammalian tissue and are essential for a wide variety of body functions. In the mammalian central nervous system, activation of purinergic receptors is involved in cellular differentiation, neurotransmission, and ion transport through ion channels [21,22], which are the gatekeepers of neuronal excitability. Interestingly, circadian changes in the gating properties of ion channels control cellular signaling mechanisms that regulate circadian gene expression and cell function [23].

However, the role of purinergic signaling within the mammalian circadian system remains largely unknown. Therefore, in this review, we focus on purinergic signaling in components of the circadian system including the retina, the SCN, and a peripheral oscillator (e.g., the urinary bladder).

## 4. Purinergic Signaling within the Retina

The mammalian retina does not only process light to generate an image of the environment but also measures environmental irradiance. The latter function is essential for light entrainment of circadian rhythms.

Light entrainment of circadian rhythms does not require functional rods and cones [24]. In 2002, a new subset of retinal ganglion cells (RGCs) was identified to be intrinsically photosensitive (ipRGCs) [25,26]. These ipRGCs express the photopigment melanopsin [25] (Figure 3). The ipRGCs do not only mediate light entrainment of the circadian system but also other non-image-forming visual functions such as the pupillary light reflex [27]. The ipRGCs project directly to the SCN and other non-image-forming brain regions such as the olivary pretectal nucleus, which controls the pupillary light reflex, the ventral sub-paraventricular zone, and the sleep active ventrolateral preoptic nucleus as well as image-forming brain regions such as the lateral geniculate nucleus [28,29]. The ipRGCs represent a very small subset (1%) of the RGCs [30] and comprises at least five subtypes (M1–M5) with differences in morphology, physiological properties, and projections (reviewed in [31]). In the retina, they form a functional gap junction-coupled unit with non-photosensitive cells [32]. Moreover, ipRGCs receive input from rods and cones that modulate their light response [33] and might account for the residual light entrainment in melanopsin-deficient mice [34,35]. Both the retinal pigment epithelium (PE) and Müller glia play roles in melanopsin chromophore regeneration [36].

The retina has its own circadian clock with a robust circadian rhythm in clock gene expression and a rhythmic release of melatonin independent of the SCN [37,38]. The retinal circadian clock and the hormone melatonin play important roles in the regulation of retinal development [39] and function [40,41,42]. Specifically, melatonin modulates the amplitude of the a- and b-waves of the electroretinogram (ERG), indicating a role of the hormone in regulating circadian changes in retinal function. The daily and circadian rhythms in the ERG response are mediated through the G-protein-coupled MT1 and MT2 receptors [43,44,45] and are PKC-dependent [45]. In addition, melatonin has receptor-independent functions including detoxification of reactive oxygen species (ROS) and other reactive molecules [13], which support the integrity of the mitochondria as well as cell function and survival [46,47]. Increased ROS production is involved in retinal pathologies such as age-related macular degeneration, glaucoma, and retinopathy [13]. Thus, melatonin is a potential preventive and therapeutic agent in the treatment of these diseases [13].

Interestingly, release and content of retinal adenosine are higher at night as compared to day, suggesting regulation by a circadian clock [48]. This darkness-evoked increase in the level of extracellular adenosine results primarily from an increase in the conversion of extracellular ATP into adenosine [48]. In addition, light leads to a decrease in extracellular adenosine levels presumably because of decreased ATP release [48]. Possible sources of extracellular ATP in the retina include both neurons as well as Müller glia [49]. Activity of ecto-ATPase could be demonstrated in the IPL surrounding Müller glia processes and rod bipolar cell terminals [50]. Ectonucleotidases hydrolyze ATP to ADP and ultimately to adenosine [51] and, therefore, play a pivotal role in purinergic signal transmission as they control their availability at purinergic P2 receptors [52]. In various prosencephalic brain regions, ectonucleotidases show a time-of-day-dependent expression pattern, which is modulated by melatonin [53], and is mediated by MT1 and MT2 receptors [54] suggesting the melatoninergic signaling as an interface between the purinergic system and the circadian system in general [54]. However, little is known about the role of melatonin on the purinergic system in the retina.

Purines can contribute to retinal neurotransmission and/or neuromodulation (reviewed in [55]). At the mRNA level, ionotropic P2X receptors [56,57,58,59,60,61,62] and metabotropic P2Y receptors [58,59,63] are expressed on RGCs, bipolar cells, and Müller glia. Using immunohistochemistry, the P2X and P2Y receptor subtypes could be detected in all layers and cell types of the retina. Further characterization of the P2X receptors by colocalization studies using immunofluorescence and/or electron microscopy revealed that P2X receptors are segregated to specific circuits within the retina [55]. The P2X_2_ receptor could be detected on a subpopulation of GABAergic (but not dopaminergic) amacrine cells, on large RGCs (type 1 alpha), and within the IPL associated with cone bipolar cell axon terminals [64]. The P2X3 receptor is also present on GABAergic amacrine cells but within the INL associated with both the rod and cone bipolar cell terminals [50]. The P2X7 receptor could be detected in the outer plexiform layer (OPL) at the rod and cone photoreceptor terminals and the horizontal cell processes as well as in the inner plexiform layer (IPL) associated with rod bipolar cell axon terminals [65] (Figure 4).

Functionally, P2X7 signaling modulates the a-wave of the ERG [65] and RGC response to a light stimulus [66], consistent with its putative role in modulating photoreceptor and RGC function. Moreover, ATP induces apoptosis of photoreceptors presumably by P2X7 signaling [67]. In the absence of extracellular ATP, P2X7 receptors expressed on macrophages can act as scavenger receptors that play an important role in the innate immune system [68]. There is increasing evidence that P2X7 receptor function contributes to retinal pathologies such as glaucoma [69,70,71], inherited retinal degenerations, as well as inflammatory changes associated with AMD [72] and diabetic retinopathy (reviewed in [68]). However, little is known about the interaction of the circadian clock and/or melatonin and the purinergic system in retinal physiology and pathology.

## 5. Purinergic Signaling within the SCN

The SCN is the central circadian oscillator that coordinates circadian rhythms throughout the body (Figure 1 and Figure 2). SCN neurons show circadian rhythms in electrical (reviewed in [73]) and metabolic (reviewed in [74]) activity as well as neurotransmitter release (reviewed in [73]). At the cellular level, the molecular clockwork in the SCN is self-sustained. However, intercellular network properties are essential for a coherent and strong circadian rhythmicity (reviewed in [75]). SCN neurons are primarily GABAergic [76] but corelease of a variety of neuropeptides [77]. Two major functionally different subdivisions of the SCN are the VIPergic ventrolateral core region and the AVPergic dorsomedial shell region [78] (Figure 5).

The core region receives direct input from the retinal ganglion cells, while the shell region sends rhythmic signals to other brain regions to synchronize the other brain and peripheral clocks. However, the core region is essential also for rhythmic behavior [79]. Rhythmic clock gene expression in the core region depends upon the light/dark cycle, while those in the shell region persist even in constant darkness [80]. VIP and GABA are crucial for coupling within the SCN network, while VIP and AVP are important for coupling the SCN with other brain regions (reviewed by [81]). There is increasing evidence that communication between astrocytes and neurons contributes largely to circadian rhythm generation [82,83]. However, mice with a targeted deletion of the glial gap-junction proteins connexin30 and connexin43 show a relatively mild circadian phenotype [84].

ATP plays a significant role in astrocyte–astrocyte as well as astrocyte–neuron intercellular communication [85,86]. Many brain regions exhibit a circadian rhythm in ATP content that is negatively correlated with electrical and metabolic activity [87]. Among these, the amplitude of the ATP rhythm is the largest in the SCN [88]. Moreover, in the SCN the rhythm of extracellular ATP accumulation persists in vitro [89] and is dependent on clock gene expression and inositol triphosphate signaling [90]. Thus, extracellular ATP might play an important role in intercellular communication within the SCN. Moreover, ectonucleotidases show a circadian rhythm in various brain regions [54]. They rapidly hydrolyze extracellular ATP to its metabolites: ADP, AMP, and adenosine [91]. This process terminates P2 activation and prevents receptor desensitization [92]. However, ADP is also an agonist for the Gi-coupled P2Y12 receptor and the Gq-coupled P2Y1 receptors [93]. Thus, both extracellular ATP content and ectonucleotidase activity determine P2 receptor agonist availability. Various P2X and P2Y transcripts [94,95] and proteins [94,96] are expressed in the SCN. P2X7 and P2Y receptors contribute to the ATP-stimulated increase of intracellular calcium levels in SCN cells [94]. Moreover, P2X2 receptor signaling modulates GABAergic inhibitory synaptic transmission in the SCN [94]. These data suggest a role of ATP and purinergic receptor signaling in SCN synaptic transmission. Rhythmic receptor activation can be achieved by rhythmic availability of the agonist and/or of the receptor. Therefore, we have previously analyzed time-of-day-dependent expression of P2 receptors in the SCN [97]. At the mRNA level, *P2X2, P2X3, P2X4, P2X5, P2X7, P2Y1, P2Y4, P2Y6, P2Y12,* and *P2Y14* show a time-of-day-dependent variation with lowest levels at the end of the dark phase and highest levels at the early dark phase [97]. At the protein level, P2X1, P2X3, and P2X4 as well as P2Y2, P2Y6, P2Y12, and P2Y14 show a time-of-day-dependent oscillation (Table 1). P2X4 has the strongest expression in the SCN (Figure 6). P2X3 is increased during the late light phase in the core region, and P2X1, P2X3, as well as P2X4 have the highest levels during the dark phase in both SCN subregions (Table 1). In contrast, P2Y12 is increased during the late dark phase in the core region, and P2Y2, P2Y6, P2Y12, as well as P2Y14 have highest levels during the early and mid-light phase in both SCN subregions (Table 1). These data show a temporal and spatial redistribution of P2 receptor subtypes in the SCN and suggest a potential role of P2 signaling in light entrainment and coupling between the SCN subdivisions. P2X receptors are more abundant during darkness and are, thus, in phase with the highest levels of their dominant agonist, ATP [89]. In contrast, P2Y receptors are more abundant during the light phase and are, thus, in antiphase with the ATP peak. This is consistent with P2Y receptor activation by other agonists besides ATP such as ADP or pyrimidines. However, little is known about the rhythmicity of extracellular ADP of other purines or pyrimidines in the SCN.

## 6. Purinergic Signaling within a Peripheral Oscillator, the Urinary Bladder

As mentioned above, the circadian system controls rhythmic behavior and physiology to ensure optimal performance. Behavior and physiology are tightly locked to the sleep–wake cycle. Diurnal animals are not only physically and mentally more active during the day but also consume and metabolize most of their food and water during this time, whereas nocturnal animals do so during the night. Thus, for any study of the behavior or physiological function, the subjective time of day of the experimental animal should be taken into consideration. In this context, it is important to emphasize that laboratory rodents such as mice and rats are nocturnal and, thus, are not in the same phase as the diurnal human. However, in both diurnal and nocturnal animals, the SCN provides endocrine and neuronal rhythmic signals for temporal synchronization within the body. The SCN directly targets the hypothalamic center for hormonal and autonomic control, the paraventricular nucleus (PVN) [98]. The PVN contains 1) neuroendocrine neurons controlling the pituitary hormone secretion and 2) preautonomic neurons. The preautonomic PVN neurons balance sympathetic and parasympathetic drive to the organs by projections to the intermediolateral column of the spinal cord and the dorsal motor nucleus of the vagus, respectively [17]. Purines act as cotransmitters with acetylcholine in parasympathetic nerves acting on P2X receptors in many different organs [3]. However, little is known about the rhythmic release of ATP and rhythmic activation of purinergic receptors in the periphery. Interestingly, purinergic signaling in the urinary bladder has been known since the 1970s (reviewed in [2,99]). On the other hand, urinary bladder function is highly rhythmic [100]. Therefore, this section focuses on the current knowledge about rhythmic purinergic signaling in this organ.

The muscles controlling micturition are innervated by autonomic and somatic nerves. During the urine storage phase, sympathetic stimulation prevails, and the internal urethral sphincter is tonically contracted while the detrusor muscle is relaxed. At increasing bladder volume, the firing rate of sensory fibers from the bladder increases, initiating the voiding reflex and causing a conscious sensation of urinary urge. During micturition, parasympathetic stimulation causes the detrusor muscle to contract and the internal urethral sphincter to relax, while somatic innervation causes the external urethral sphincter to relax. After voiding, the storage phase restarts [101,102].

Urine production and voiding occur predominantly during the active phase, whereas during the inactive phase, kidney function is decreased and the storage capacity of the urinary bladder is increased [103]. Disruption of this temporal regulation, for example, in elderly people or in patients with neurodegenerative diseases with nocturia, affects sleep quality as well as general quality of life and ultimately increases morbidity and mortality [71,72,73,74]. Therefore, a better understanding of rhythmic function of the urinary bladder is highly relevant.

In mice, the rhythm of functional bladder capacity is dependent on functional molecular clockwork [104]. Bladder smooth muscle cells have an internal molecular clockwork that drives rhythmic expression of connexin43 (Cx43), a gap junction protein [104]. Surprisingly, mice expressing a reduced level of Cx43 have a larger functional bladder capacity [104]. Thus, Cx43 negatively regulates capacity contributing to rhythmic bladder function. Consistent with its regulatory function, Cx43 is expressed in human and mouse urothelia lining the bladder lumen [105]. Furthermore, the mouse urothelia contains a molecular clockwork and displays a rhythmic Cx43 expression with highest levels during the active phase [105].

Importantly, ATP concentration in the bladder lumen undergoes daily variations, peaking in phase with Cx43. Moreover, Cx43 controls rhythmic as well as mechanically induced ATP release from urothelial cells, presumably by forming ATP-releasing hemichannels [105]. Thus, a rhythmic molecular clockwork in the urothelia might control Cx43 hemichannel function providing rhythmic ATP release. This time-of-day-dependent change in ATP release from the urothelia might shape homeostatic regulation of bladder function.

Homeostatically, increased hydrostatic pressure as well as mechanosensory stimulation, such as distension, evokes ATP release from the urothelia [106,107]. The urothelia itself expresses P2X receptors, which modulate the apical membrane composition of umbrella cells [108,109,110,111,112,113]. Suburothelial sensory nerve fibers express P2X and P2Y receptors, which respond to ATP release from the urothelial cells during bladder distension, mediating the voiding reflex and nociception [114,115,116]. Moreover, ATP acts as an excitatory cotransmitter with acetylcholine in parasympathetic nerves on P2X receptors and mediates the contractile response of the detrusor muscle [112,117,118,119,120,121,122,123]. As mentioned above, rhythmic parasympathetic innervation is controlled by the circadian system. Thus, ATP release from the parasympathetic nerve endings and consequently P2 receptor function in the detrusor muscle might undergo time-of-day-dependent changes.

We performed a mapping of P2X and P2Y receptors (Figure 7, Table 2) in the urinary bladders of mice sacrificed during the early day/inactive phase when storage capacity is increasing. In the urothelium, we found a more or less intense immunoreaction of all P2X and P2Y receptors, suggesting a potential role of all receptors in mediating autocrine/paracrine regulation of urothelial function by purines and pyrimidines (unpublished data). In the suburothelial layer, a substantial P2X1-, P2X2-, P2X7-, and P2Y1 immunoreaction was present, suggesting a major role of these receptors in mediating mechanosensation. In the detrusor muscle, a substantial immunoreaction was observed for P2X6, P2Y1, and P2Y6 (Table 2). The P2Y6 receptor is selectively activated by pyrimidines, and the contractile effect of UDP on vascular smooth muscle cells is lost in P2Y6-deficient mice [124]. This mapping of P2 receptor subtypes in the bladder provides a structural basis for further functional, pathological, and pharmacological analyses.

Importantly, under pathological conditions rhythmic bladder function is disrupted, and, as mentioned above, dysregulation of rhythmic bladder function affects the sleep–wake cycle. ATP release [123,125] as well as expression of P2X and/or P2Y receptors are changed in various bladder diseases [126,127]. Importantly, purinergic signaling has a high therapeutic potential [128]. Thus, timed application of receptor-selective purinergic drugs might be a successful therapeutic strategy for interrupting the vicious cycle of chronodisruption and bladder dysfunction.

## 7. Summary

In this review, we gathered current knowledge on purinergic signaling in the major components of the mammalian circadian system: the retina (input), the SCN (central oscillator), and the urinary bladder as an example of a peripheral oscillator (Figure 8). For the urinary bladder, we discussed the interaction of chronodisruption and organ dysfunction. However, further detailed analyses of spatiotemporal distribution of agonists and receptors are needed to better understand purinergic signaling in health and disease and to fully exploit its high therapeutic potential.

## Figures and Tables

**Figure 1 ijms-21-03423-f001:**
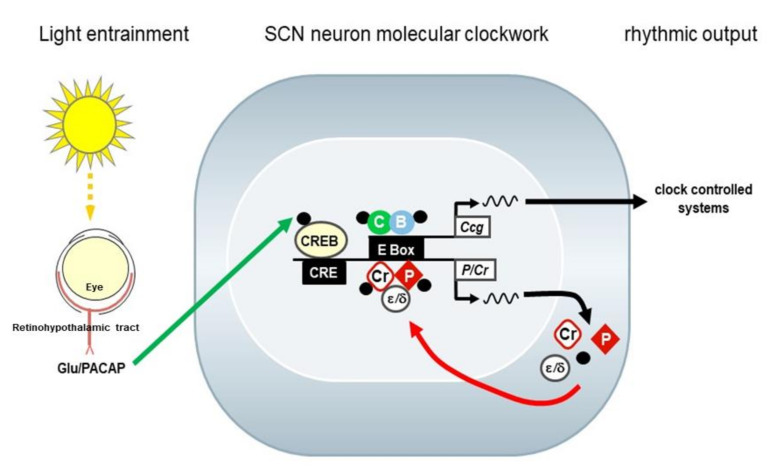
The mammalian suprachiasmatic nucleus (SCN) molecular clockwork. Light is received by the eye and transmitted by the retinal ganglion cells, forming the retino-hypothalamic tract, to the suprachiasmatic nucleus (SCN). Light/dark information is encoded by the release of the neurotransmitters glutamate (Glu) and pituitary adenylate cyclase-activating peptide (PACAP). Glutamate and PACP signal transduction lead to the activation of the transcription factor CREB, which modulates the intrinsic SCN molecular clockwork. The molecular clockwork consists of the transcriptional/translational feedback loops of so-called clock genes. The transcription factors Clock (C) and Bmal1 (B) bind to E-box elements in the regulatory region of the negative transcription regulators, the periods (P) and cryptochromes (Cr), as well as of clock-controlled genes (Ccg). The P and Cr accumulate, become phosphorylated by casein kinases (ε/δ), and form a complex that negatively interferes with C and B activity. Eventually, Cr and P transcription is downregulated, the complex is degraded, and a new cycle starts. Each cycle takes approximately 24 h (circadian). Accordingly, the Ccg are rhythmically expressed and encode for neuropeptides that transmit the time information to other parts of the hypothalamus regulating the autonomic nervous system or the endocrine system.

**Figure 2 ijms-21-03423-f002:**
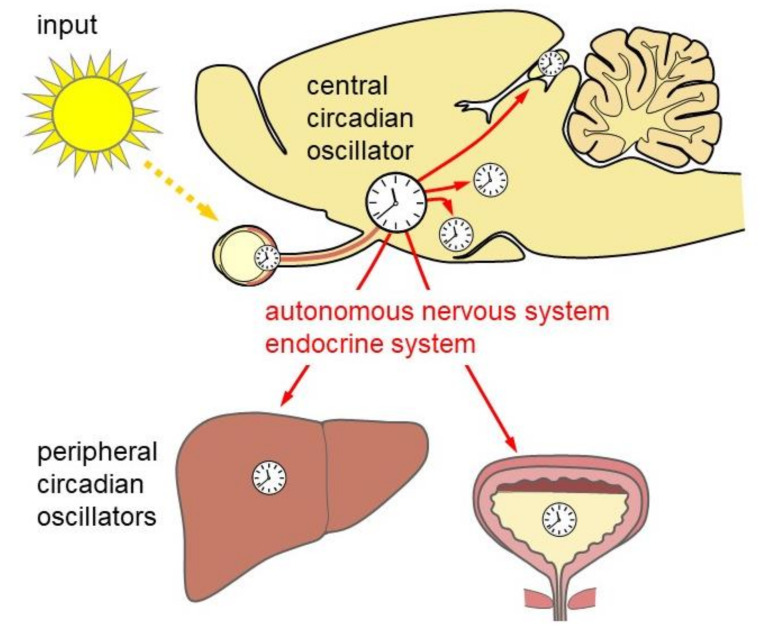
The mammalian circadian system. Light is received by the eye (input) and transmitted to the central circadian oscillator in the suprachiasmatic nucleus. The central circadian oscillator provides rhythmic output signals via the autonomous nervous system and the endocrine system to peripheral circadian oscillators in other parts of the brain and the body such as the liver (left) and the urinary bladder (right).

**Figure 3 ijms-21-03423-f003:**
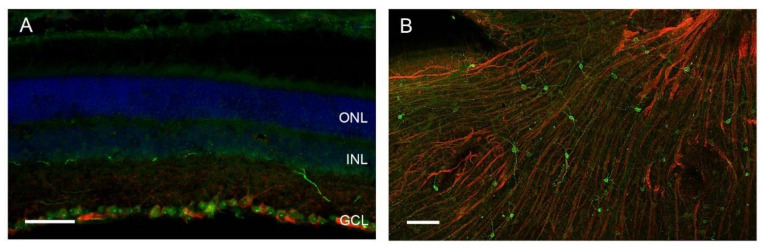
Representative melanopsin immunoreaction (green) in retinal ganglion cells (**A**) in a coronal section, scale bar 50 µm, and (**B**) in a whole-mount mouse retina preparation, scale bar 100 µm. Beta-tubulin-immunoreactive axons are shown in red, cell nuclei (DAPI) are shown in blue. ONL, outer nuclear layer; INL, inner nuclear layer; GCL, ganglion cell layer.

**Figure 4 ijms-21-03423-f004:**
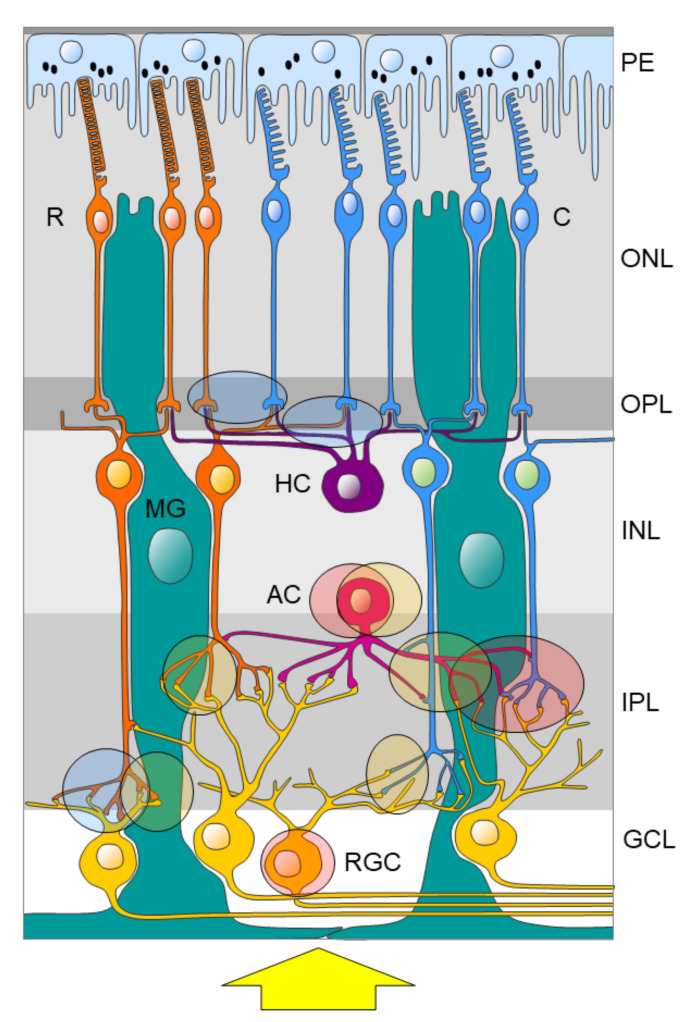
Potential function of P2X2, P2X3, and P2X7 receptors in the retina based on subcellular localization. P2X_2_ receptor activation (red-shaded form) might modulate amacrine cell (AC) as well as retinal ganglion cell (RGC) function and specifically the cone (C) signaling pathway. P2X3 receptor activation (yellow-shaded form) might modulate amacrine (AC) cell function and both the rod (R) and cone (C) signaling pathways. P2X7 receptor activation (blue-shaded form) might modulate the photoreceptor as well as horizontal cell function (HC) and specifically the rod (R) signaling pathway. The yellow arrow indicates the light direction. GCL; ganglion cell layer, INL; inner nuclear layer, IPL; inner plexiform layer, ONL; outer nuclear layer, OPL; outer plexiform layer, MG; Müller glia, PE; pigment epithelium. Subcellular localization of P2 receptors is based on previous findings [50,64,65].

**Figure 5 ijms-21-03423-f005:**
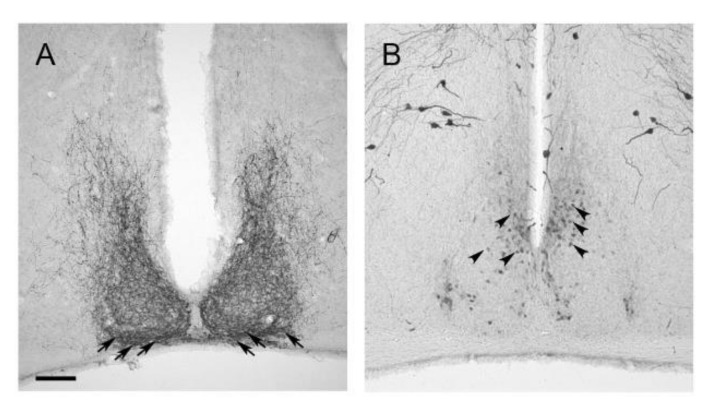
Representative immunoreaction of neuropeptides in the SCN. (**A**) VIP immunoreaction is present in perikarya of the ventrolateral core region (arrows) and a dense neuropil throughout the entire SCN. (**B**) AVP immunoreaction is present in perikarya of the dorsomedial shell region (arrowheads). Scale bar, 100 µm.

**Figure 6 ijms-21-03423-f006:**
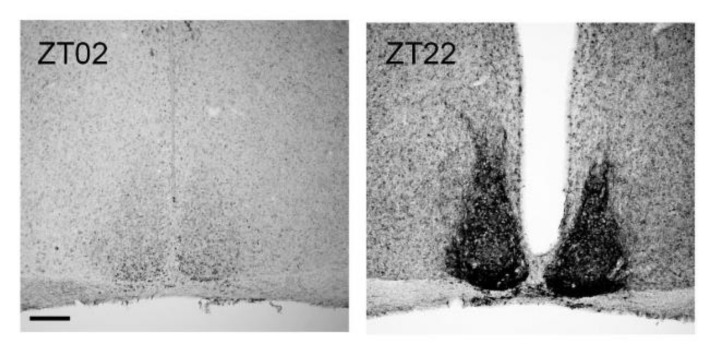
Representative P2X4 immunoreaction in the SCN. Brain sections obtained from mice at two different time points during the 12 h light/12 h dark cycle. Zeitgeber time (ZT02) is two hours after lights on, ZT22 is two hours before lights on. Scale bar, 100 µm.

**Figure 7 ijms-21-03423-f007:**
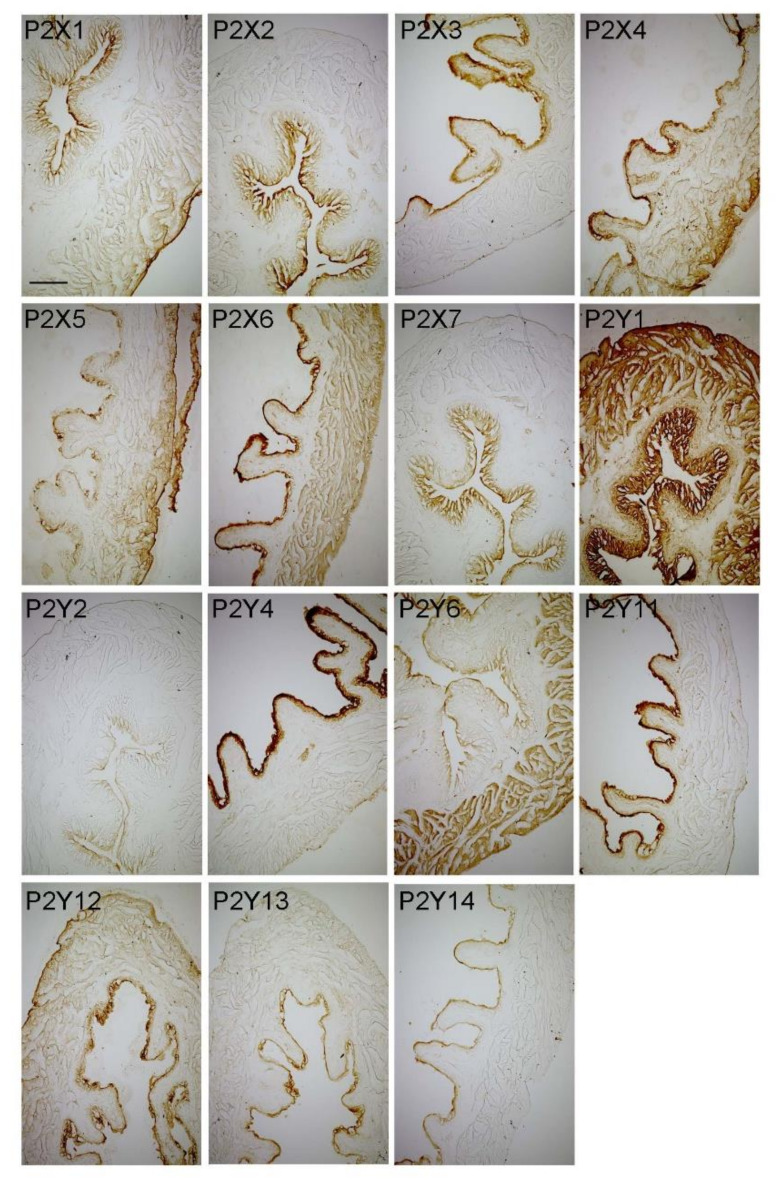
Representative microphotographs showing immunoreaction of P2X1-7 receptors and P2Y1-4, 6, and 11-14 in the mouse urinary bladder. Scale bar, 200 µm.

**Figure 8 ijms-21-03423-f008:**
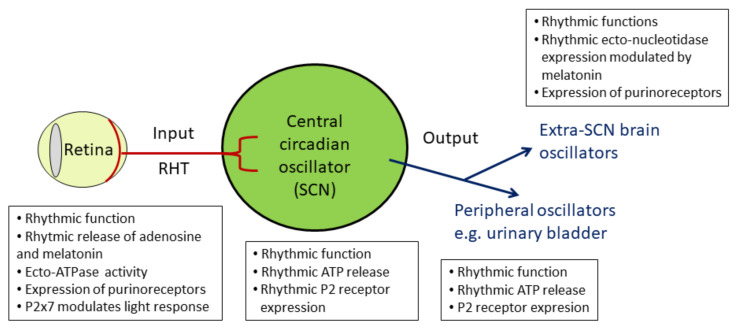
Summary of purinergic signaling and circadian rhythm in the components of the mammalian circadian system. RHT: retinohypothalamic tract, SCN: suprachiasmatic nucleus.

**Table 1 ijms-21-03423-t001:** Time-of-day-dependent expression of P2 receptors in the SCN obtained from mice at different zeitgeber times (ZTs). ZT00 is defined as lights on. White and black bars indicate light and darkness, respectively. P2 receptors with a time-of-day-dependent variation in expression are marked with an asterisk, and peak levels are indicated with bold symbols. The intensity of immunoreaction was categorized arbitrarily: + = low, ++ = moderate, +++ = high. Data are based on [97].

Zeitgeber Time (ZT)	02		06		10		14		18		22	
Light/dark												
subregion	core	shell	core	shell	core	shell	core	shell	core	shell	core	shell
P2X1 *	+	+	+	+	+	+	+	+	+	+	**++**	**++**
P2X2	+	+	+	+	+	+	+	+	+	+	+	+
P2X3 *	+	+	+	+	**++**	+	**++**	**++**	**++**	**++**	**++**	**++**
P2X4 *	++	++	++	++	++	++	**+++**	**+++**	**+++**	**+++**	**+++**	**+++**
P2X5	++	++	++	++	++	++	++	++	++	++	++	++
P2X6	++	++	++	++	++	++	++	++	++	++	++	++
P2X7	++	++	++	++	++	++	++	++	++	++	++	++
P2Y1	+	+	+	+	+	+	+	+	+	+	+	+
P2Y2 *	**++**	**++**	**++**	**++**	+	+	+	+	+	+	+	+
P2Y4	+	+	+	+	+	+	+	+	+	+	+	+
P2Y6 *	+	+	**++**	**++**	+	+	+	+	+	+	+	+
P2Y11	+	+	+	+	+	+	+	+	+	+	+	+
P2Y12 *	**++**	**++**	+	+	+	+	+	+	+	+	**++**	+
P2Y13	++	++	++	++	++	++	++	++	++	++	++	++
P2Y14 *	**++**	+	**++**	**++**	+	+	+	+	+	+	+	+

**Table 2 ijms-21-03423-t002:** Expression of P2 receptors in the urinary bladder wall subregions. The intensity of immunoreaction was categorized arbitrarily: -, absent; (+), very weak; +, low; ++, moderate; +++, high. Immunohistochemistry was performed as described previously [97].

P2 Receptors	Urothelium	Sub-Urothelium	Detrusor Muscle
P2X1	++	++	(+)
P2X2	++	++	-
P2X3	++	-	-
P2X4	+++	-	+
P2X5	++	-	+
P2X6	+++	-	++
P2X7	++	++	-
P2Y1	+++	+++	+++
P2Y2	+	+	-
P2Y4	+++	-	-
P2Y6	+	-	++
P2Y11	+++	-	-
P2Y12	++	-	+
P2Y13	+	-	-
P2Y14	+	-	-

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
