# Peer review of "The Role of Purinergic Receptors in the Circadian System"

_ijms, 2020, doi:10.3390/ijms21103423_

Round 1

Reviewer 1 Report

In the present review, Amira AH Ali and colleagues discuss the molecular mechanisms involved in the regulation of the circadian rhythm with a specific focus on the interaction with the purinergic system. First, the authors describe the circadian rhythm and the molecular clock. Then, they discuss the role of purinergic receptors in the retina and in the suprachiasmatic nucleus (SCN) as well as in peripheral tissues. This last part allows to highlight the link between disruption of the molecular clock and organ dysfunction, notably in the bladder. This part is certainly interesting for a broad audience as it underlines why the interaction between the 2 systems might be important.

The authors covered well the fields of circadian clock and purinergic systems (123 references). This group has obviously an expertise in the field, as they published several articles on the circadian rhythm and recently an article specifically on the interaction between purinergic receptor expression and circadian clock in the suprachiasmatic nucleus (Lommen J et al., 2017).

In conclusion, the review is clear, well written, well documented and the illustrations in figure 1, 2 and 5 are excellent. I think this review can be interesting for a broad audience and therefore deserves to be published in a good journal such as IJMS.

Minor comments:

Since purinergic receptors mediate ion flux into the cells, it would have been nice to mention that the interaction between circadian rhythm and ion channels will be important to fully understand the link between circadian clock and purinergic system, notably in the context of channelopathies.

The table 1 and the Figure 5 could probably be combined into a single figure to improve the readability.

Typos

L47 “formtogether” should be spelled “form together”

L47 there is a double space between “kinases” and “a”

Author Response

Reviewer 1:

Minor comments:

  1. Since purinergic receptors mediate ion flux into the cells, it would have been nice to mention that the interaction between circadian rhythm and ion channels will be important to fully understand the link between circadian clock and purinergic system, notably in the context of channelopathies.

Response: we have discussed this issue in section 3.

  1. The table 1 and the Figure 5 could probably be combined into a single figure to improve the readability.

Response: we skipped the table as the figure 5 focused on the most prominent P2 receptors.

  1. Typos

L47 “formtogether” should be spelled “form together”

L47 there is a double space between “kinases” and “a”

Response: we have corrected the typos.

Reviewer 2 Report

The authors study role of purinergic receptors in circadian signalling, preferentially in the SCN. They provided a decent text. I have several objections that should be corrected before publishing of MS.

Major comments:

Statement for chapter 2 that “Circadian system consists of three major components: a central circadian oscillator, input pathways to allow entrainment and output pathways that regulate overt circadian rhythms in behaviour and physiology.” is not correct. The authors must consider also peripheral oscillator if they intend to make statement about the circadian system and not only about master clock. This must be corrected.

- the main function of casein kinase (that is an enzyme) in relation to clockwork is to facilitate per degradation, in this respect a newer reference should be mentioned together with Reppert and Weaver 2002 and information added into text in page 2

paragraph: “The muscles controlling micturition are innervated by autonomic and somatic nerves. During the urine storage phase, sympathetic stimulation prevails and the internal urethral sphincter is tonically contracted while the detrusor muscle is relaxed. At increasing bladder volume, firing rate of sensory fibres from the bladder increases, initiating the voiding reflex and causing a conscious sensation of urinary urge. During micturition, parasympathetic stimulation causes the detrusor muscle to contract and the internal urethral sphincter to relax while somatic innervation causes the external urethral sphincter to relax. After voiding, the storage phase restarts.”

misses references, they should be added.

Text describing purinergic signalling is very descriptive. Since the authors submitted a scientific review and not a textbook, there is no reason for publishing Figure 3. In steady of it they should provide scheme of explaining purinergic interaction with basic circadian loop and/or circadian system.

A function of chapter 4 is not clear. Do the authors describe interaction of purinergic signalling with retina as an example of modified input pathway or as an example of peripheral oscillator? Or with both? This should be clearly stated.

In chapter 6 the authors refer only about urinary bladder function. The name of the article and chapter should be changed accordingly or involve also other peripheral oscillators.

Author Response

Reviewer 2:

Major comments:

  1. Statement for chapter 2 that “Circadian system consists of three major components: a central circadian oscillator, input pathways to allow entrainment and output pathways that regulate overt circadian rhythms in behaviour and physiology.” is not correct. The authors must consider also peripheral oscillator if they intend to make statement about the circadian system and not only about master clock. This must be corrected.

Response: The cited statement is about circadian systems in general. In some organisms, the circadian system combined in a single cell. The peripheral oscillator is more specific to the mammalian circadian system, which is introduced, in a later paragraph. We tried to make this more comprehensible.

  1. The main function of casein kinase (that is an enzyme) in relation to clockwork is to facilitate per degradation, in this respect a newer reference should be mentioned together with Reppert and Weaver 2002 and information added into text in page 2

Response: this has been adjusted.

  1. Paragraph: “The muscles controlling micturition are innervated by autonomic and somatic nerves. During the urine storage phase, sympathetic stimulation prevails and the internal urethral sphincter is tonically contracted while the detrusor muscle is relaxed. At increasing bladder volume, firing rate of sensory fibres from the bladder increases, initiating the voiding reflex and causing a conscious sensation of urinary urge. During micturition, parasympathetic stimulation causes the detrusor muscle to contract and the internal urethral sphincter to relax while somatic innervation causes the external urethral sphincter to relax. After voiding, the storage phase restarts.”misses references, they should be added.

Response: This is textbook knowledge, however references have been added.

  1. Text describing purinergic signalling is very descriptive. Since the authors submitted a scientific review and not a textbook, there is no reason for publishing Figure 3. In steady of it they should provide scheme of explaining purinergic interaction with basic circadian loop and/or circadian system.

Response: figure 3 has been deleted and additional scheme explaining the interaction has been added as figure 8.

  1. A function of chapter 4 is not clear. Do the authors describe interaction of purinergic signalling with retina as an example of modified input pathway or as an example of peripheral oscillator? Or with both? This should be clearly stated.

Response: this has been stated clearly in chapter 4.

  1. In chapter 6 the authors refer only about urinary bladder function. The name of the article and chapter should be changed accordingly or involve also other peripheral oscillators.

Response: the name of the chapter has been adjusted.

Round 2

Reviewer 2 Report

Changes that have been made are mostly formal (except of Figure 8). In spite of the fact that the MS does not have a strong concept, it can be published to provide a wide range of detailed information covering not very often studied topic. The authors do use term “circadian system” inappropriately. This is a review about SCN, retina and urinary bladder, not about a circadian system. My attempt to improve concept was not successful. However, it should be done.